# Assessment of Sustainable Supply Capability of Chinese Tin Resources Based on the Entropy Weight-TOPSIS Model

**Xiaolei Fan [1], Hao Li [1], Qinfei Yu [2,\*], Jianming Xu [3] and Meng Li [4]**

[1] Faculty of Earth Resource, China University of Geosciences, Wuhan 430074, China; fanxiaolei@cug.edu.cn (X.F.); severus.lee@cug.edu.cn (H.L.)

[2] Chinese Academy of Natural Resources Economics, Beijing 101149, China

[3] School of Earth Sciences and Resources, China University of Geosciences (Beijing), Beijing 100083, China; jianmxu@email.cugb.edu.cn

[4] China Association for Geological Disaster Prevention and Ecological Restoration, Beijing 100043, China; limeng01@cug.edu.cn

\* Correspondence: qfyu@canre.org.cn

**Abstract:** This study focuses on the issue of supply security in Chinese tin resources, conducting a comprehensive analysis encompassing supply security, demand conditions, market risk, and geopolitical stability. The study employs the AHP-Entropy Weight method to calculate the comprehensive weights of various indicators, revealing the sustainable supply capability of the resources and the challenges they confront. The research unveils that the domestic supply potential and the international commercial extraction environment are pivotal influences on the sustainable supply capability of tin resources. Firstly, since 2010, China has started to import net tin ores and concentrates, leading to a substantial increase in external dependency. However, China's tin resource reserves have gradually decreased, posing challenges to the stability of the tin resource supply. Secondly, the stability of supply countries and the security of import channels critically affect the stability of the tin resource supply chain. A comprehensive analysis of the outcomes indicates a consistent decline in China's sustainable tin resource supply capability since 2010, primarily due to the deterioration of resource supply conditions, growth in demand, and the escalation of geopolitical risks. Thus, the evolving foreign extraction environment, domestic resource supply potential, and the alteration in the global tin resource extraction landscape require significant attention.

**Keywords:** tin mineral resources; sustainable supply capacity; entropy Weight-TOPSIS model; supply security; resource conservation





## 1. Introduction

Tin, as a strategic mineral resource, holds a pivotal role across numerous industries, including electronics, construction, packaging, and transportation manufacturing. China stands as the world's leading nation in terms of tin reserves, production, and consumption. As strategic emerging industries continue to flourish, the demand for tin ores is projected to maintain a growth trajectory. Since 2010, China has transitioned from being a net exporter of tin resources to becoming a net importer. By 2015, the external dependency on tin concentrates had surpassed the 50.0% mark [1]. Over the past several decades, China's tin resource sector has undergone rapid exploitation and overutilization, leading to the emergence of threats such as resource depletion, environmental degradation, and social issues in various mining areas. This, in turn, has resulted in an inadequate self-sufficiency of tin resources within the country, amplifying China's reliance on international imports [2,3]. Nevertheless, potential factors such as trade disputes, geopolitical tensions, and fluctuations in international policy may impinge upon China's tin resource supply chain, exerting an influence on the nation's tin resource provisioning. Consequently, the

evaluation of China's sustainable capability in the realm of tin resource development emerges as an exigent undertaking.

The study of the sustainable supply capacity of mineral resources originated in the 1970s with global concerns about the environment and resources. During that time, researchers such as Knock, A. (1979) [4], Hennart, J. F. (1986) [5], and Levine, R. (1994) [6] began to recognize the finite nature of mineral resources and the impacts of mining activities on the environment and society. Their focus was primarily on reserve evaluation, development potential analysis, and environmental impact assessment of mineral resources. With the rise of the concept of sustainable development, the research on the sustainable supply capacity of mineral resources entered a new stage in the 21st century. In this stage, scholars like Xueshan, Y. (2005) [7] and Ishihara, S. (2008) [8] began to pay attention to the sustainable development and utilization of resources, emphasizing the balance between economic, social, and environmental benefits in resource development. The research methods gradually shifted from single-indicator evaluations to multidimensional evaluations, including comprehensive assessments of economic, environmental, and social aspects. Following the 2009 global financial crisis, the research on the sustainable supply capacity of mineral resources further deepened and improved. Researchers such as Wanli, X. I. (2016) [9], Sonderegger, T. (2020) [10], and Tomazinakis, S. (2022) [11] began to focus on more comprehensive and systematic evaluation methods, including life cycle assessment, risk assessment, and ecological footprint analysis. Moreover, studies such as Adibrata, S. (2021) [12] started to consider the impacts of mining activities on communities and local residents, emphasizing the importance of social responsibility and engagement.

Research on the assessment of the sustainable supply capacity of mineral resources encompasses various aspects, including resource evaluation and reserve prediction, global supply–demand patterns, socio-economic impact assessment, as well as sustainable development strategies and management measures [13,14]. Resource evaluation and reserve prediction serve as the foundation for assessing sustainable supply capacity by analyzing the geological characteristics, quantity, and quality of resources to determine their supply potential and extraction difficulty [15,16]. The evaluation of global supply–demand patterns focuses on major countries' supply, consumption, and demand for mineral resources worldwide, aiming to assess and predict a nation's international status regarding mineral resources over a specific period and to propose corresponding management measures [17,18]. Socio-economic impact assessment considers the influence of mining activities on the social and economic aspects, providing a comprehensive evaluation of the socio-economic benefits while emphasizing factors such as fairness, sustainable development, and social responsibility [19]. Additionally, the assessment of the sustainable supply capacity of mineral resources needs to consider factors such as resource supply potential, global resource markets, supply chain security, and geopolitical security [20]. Evaluating the resource supply potential determines the quantity and quality of resources, providing a basis for their rational development and utilization [21,22]. Analysis of the global resource market considers factors such as supply–demand relationships, price fluctuations, and international trade [23]. The assessment of supply chain security focuses on vulnerable links, risks, and reliability within the supply chain to ensure sustainable resource supply and mitigate the risk of supply disruptions [24,25]. Geopolitical security considerations involve a comprehensive analysis of political stability, legal and regulatory environments, as well as social and economic conditions in the regions where the resources are located [26].

The assessment of the sustainable supply capacity of mineral resources involves the interdisciplinary and comprehensive integration of natural sciences, social sciences, and engineering technologies. By synthesizing indicators related to resource quantity, quality, environmental impact, and socio-economic benefits, it provides a scientific basis for the sustainable development and utilization of resources, thereby promoting resource management and sustainable mining practices. Through research and evaluation of these aspects, scientific foundations can be established to facilitate the sustainable development

and utilization of resources, as well as to advance resource management and the sustainable development of the mining industry.

Overall, the research on the assessment of sustainable supply capacity of mineral resources has undergone an evolutionary process, shifting from a focus on resource reserves to the exploration of resource development and utilization methods. This evolution has seen a transition from single-indicator evaluations to multidimensional assessments and, more recently, the application of more comprehensive and systematic evaluation methods. Moving forward, future research will continue to explore more scientific and effective evaluation methods to support the development of sustainable mining practices and foster the sustainable utilization of resources, contributing to sustainable socio-economic development.

This paper aims to evaluate the sustainable supply capacity of tin mineral resources in China by deeply understanding the connotation of mineral resource security under new circumstances. By combining the endowment of tin mineral resources in China and the characteristics of industrial development, AHP-Entropy Weight is employed to construct an evaluation index system for the sustainable development of tin resources in China. The evolution of China's tin mineral resource sustainability from 2010 to 2022 is studied, and targeted recommendations are provided based on the specific factors constraining resource supply security. This research aims to provide a basis for the future development, utilization, and international positioning policies of China's tin mineral resources in a complex international landscape, offering valuable insights for ensuring the security of China's tin mining.

The innovations of this study are mainly reflected in the following aspects: firstly, we have constructed a comprehensive evaluation framework that takes into account supply security, demand trends, market risks, and geopolitical factors, providing a comprehensive assessment of China's sustainable tin mining supply capability. Secondly, we have employed the Analytic Hierarchy Process (AHP) in combination with the entropy weight method to determine the weight of indicators, making the evaluation more objective and systematic. Furthermore, we have conducted an in-depth analysis of the critical impact of geopolitical factors, identifying vulnerabilities brought by reliance on specific import channels, and providing valuable insights for policymakers. Lastly, we have proposed forward-looking policy recommendations to strengthen the security of China's tin mining supply, which address future trends and offer valuable guidance to cope with the ever-changing global environment.

## 2. Overview of Global Resources

Tin is a metallic element characterized by its silver-white luster, with a melting point of 23 °C. It predominantly exists in the form of ores and compounds, such as dioxide and various sulfides, and exhibits resistance to oxidation. Figure 1 illustrates the material flow diagram of the entire tin ore industry chain, encompassing the process from extraction to downstream applications. This diagram provides an overview of tin ore processing yields and consumption sectors. The pie chart within the diagram presents the consumption structure of tin ore resources in China and globally. Through this figure, various application scenarios of tin ore resources are depicted, serving as a foundation for predicting consumption demands. The data for this figure are referenced from the United States Geological Survey. Tin is a crucial material in the development of high-tech industries, including aerospace, defense, high-tech product manufacturing, and biomedicine. In terms of global tin consumption structure, it remains relatively stable. The proportion of tin consumed in tin-plated sheets shows a downward trend, while the proportion consumed in tin solder exhibits an upward trend. In 2022, the proportion of global tin solder consumption reached 49.0%, while tin chemicals and tin-plated sheets accounted for 17.6% and 12.2%, respectively. Lead–acid battery consumption accounted for 8.1%, copper alloy consumption accounted for 7.1%, and consumption in other sectors accounted for 8%. Meanwhile, China's tin mineral resources are primarily used in the electronics sector (44.6%), followed by tin-plated sheets (16.3%) and chemical products (14.1%).

The future outlook for global tin mineral resource consumption demonstrates a continued growth trend. Firstly, as the global economy continues to expand and industrialization advances, the demand for tin mineral resources is expected to increase [27]. Secondly, with the development of emerging technologies such as electric vehicles, renewable energy, and smart devices, the demand for tin mineral resources is anticipated to further rise [28]. Additionally, China, being the world's largest importer and consumer of tin mineral resources, heavily relies on foreign supply to meet domestic demand, making it a significant driver of global tin mineral resource demand growth. With sustained economic growth and technological innovation, China's consumption demand for tin mineral resources is projected to continue its upward trajectory.

The global supply of tin mineral resources faces pressures and constraints that may lead to a tense supply–demand situation. Firstly, countries such as China, Myanmar, and Indonesia have limited tin mineral reserves [29,30] and face challenges in extraction, potentially resulting in insufficient supply and price fluctuations [31]. Secondly, the requirements for environmental protection and sustainable development impose higher standards and demands on the extraction and utilization of tin mineral resources. Building secure and stable supply chains and promoting the development of circular economy practices and green technologies will become crucial directions for the future [32].

In summary, influenced by global economic development, the drive from emerging technologies and China's economic growth, both global and Chinese future prospects for tin mineral resource consumption exhibit a growth trend. However, it is essential to address concerns regarding supply–demand pressures, environmental requirements, and the challenges of sustainable development. Therefore, ensuring the sustainable development of tin mineral resources in the future becomes crucial, necessitating the strengthening of supply–demand balance, promotion of efficient resource utilization, and environmentally-friendly mining practices.

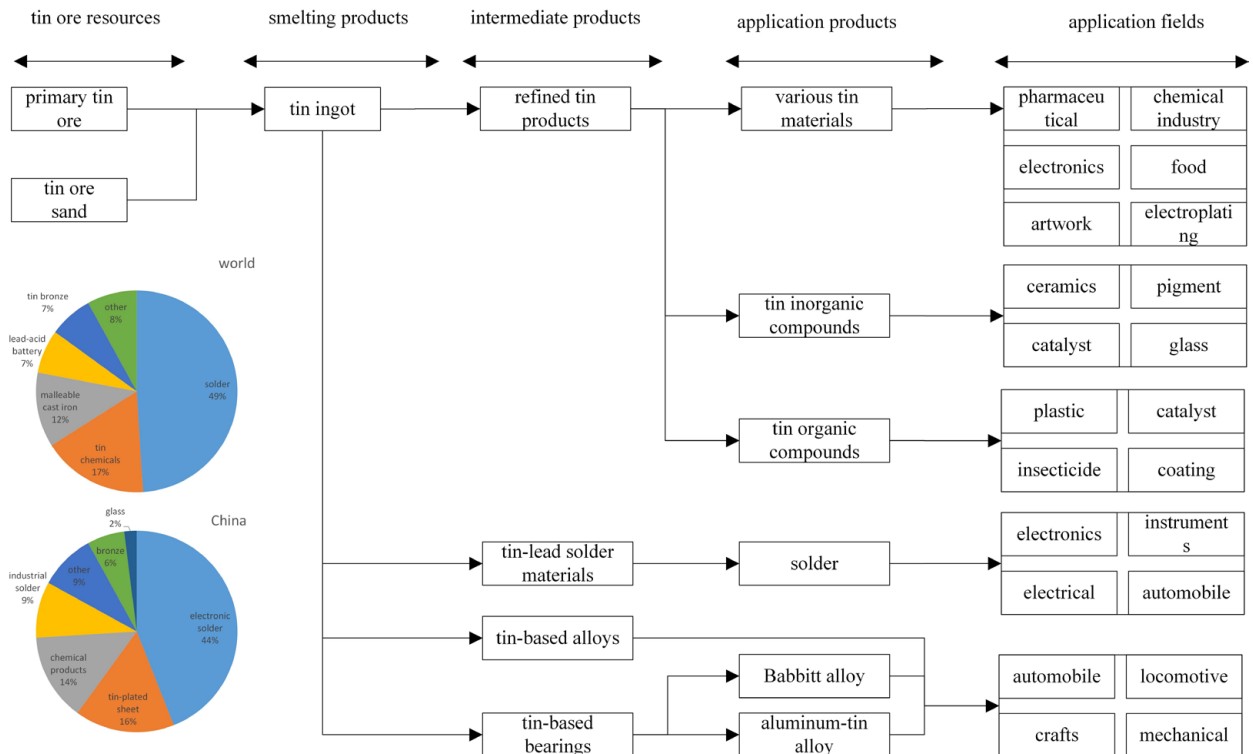

**Figure 1.** Schematic diagram of material flow of tin ore resources.

## 3. Data Sources and Research Approach

### 3.1. Indicator System

Currently, as China's economy and geopolitical landscape enter a new development stage, the development of the tin mineral resource industry faces a series of new opportunities and challenges. Firstly, China's industrial structure is continuously adjusting, with a focus on high-end manufacturing as a national development priority [33]. With the deepening of technological and industrial revolutions, the application of tin mineral resources in high-end manufacturing will play a significant role. Secondly, there is a profound adjustment in international powers, intensifying competition among major nations, and an increase in regional conflicts and instability, posing greater challenges to China's tin mineral resource development environment [34]. In international competition, China needs to pay more attention to the sustainable utilization of tin mineral resources, enhance resource supply security, and explore substitutability. Lastly, with the advancement of carbon neutrality goals, global energy structures and related industries are rapidly changing, leading to an increased demand for the substitution of traditional energy and clean energy technologies. This will have a profound impact on the development and utilization of China's tin mineral resources [35]. In this new development context, it is crucial for China to strengthen the evaluation of the sustainable capacity of tin mineral resources, promote efficient resource utilization, and pursue green development to address challenges and seize opportunities in the face of this new landscape.

The concept of sustainable capacity of mineral resources carries different meanings in different periods and among researchers. Previous studies primarily emphasized the ability to stably and continuously access mineral resources, focusing on resource control and supply. However, recent research has embraced a broader understanding of mineral resource security, incorporating aspects such as supply security, ecological security, and market security [36]. Scholars such as Asr, E. T. (2019) [37], Potravny, I. (2022) [38], and Zhang, Z. (2022) [39] have further expanded the scope of the sustainable capacity of mineral resources, including factors such as geopolitical security, military security, economic security, environmental security, technology, and efficiency. Considering various perspectives, mineral resource security encompasses factors at both domestic and international levels. At the domestic level, the main focus includes resource endowment, supply and demand dynamics, development levels, environmental impacts, economic factors, and technological factors. At the international level, factors such as international relations, resource markets, transportation security, and military security are considered.

In light of the consumption prospects for China's tin mineral resources and the current global situation [40], this study summarizes the influencing factors on the sustainable capacity of China's tin mineral resources into four aspects: (1) Supply Security, which encompasses concerns about the supply security of tin mineral resources, including domestic resource supply potential and the international market's business environment for resources; (2) Demand Situation, which refers to the growth rate of resource consumption and the degree of dependence on foreign sources, as these factors limit the security of resource consumption; (3) Market Risk, which includes the degree of resource monopolization and the extent of price fluctuations, influencing the direction of the market; and (4) Geopolitical Security, which considers the stability of resource-supplying countries and their military control over trade routes in the new context, as these factors represent significant potential risks affecting resource supply.

Table 1 presents the evaluation index system for the sustainable capacity of China's tin mineral resources and their corresponding calculation methods.

**Table 1.** Evaluation Index System for Sustainable Development of China's Tin Resources.

| Primary Indicators | Secondary Indicators | Indicator Attributes | Indicator Calculation Formula |
|---|---|---|---|
| Supply Security (*FLG*) | Business Environment (*FLG1*) | + | $FLG1 = \sum IEF_i \times \frac{P_i}{P}$<br>In the equation: *i* represents a tin-producing country. $P_i$ represents the tin production of country *i*. *P* represents the global total tin production in year *i*. $IEF_i$ represents the economic freedom index of country *i*, sourced from the Heritage Foundation. |
| | Domestic Supply Potential (*FLG2*) | + | $FLG2 = \frac{R_i}{P_i}$<br>In the equation: *i* represents a year. $R_i$ represents the tin resource reserves in China in year *i*. $P_i$ represents the tin production in China in year *i*. The data for China's tin resource reserves and production are sourced from the United States Geological Survey. |
| Demand Situation (*FLX*) | Consumption Growth Rate (*FLX1*) | - | $AP = Pr + Im - Ex$<br>$FLX1 = \frac{AP_n - AP_{n-1}}{AP_{n-1}}$<br>In the equation: *AP* represents the apparent consumption of tin. n represents a year. *Pr* represents the tin production of tin resources. *Im* represents the tin imports of tin resources. *Ex* represents the tin exports of tin resources. The production data are sourced from the United States Geological Survey (USGS). The trade data are sourced from the United Nations Commodity Trade database. |
| | External Dependency (*FLX2*) | - | $FLX2 = \frac{Im - Ex}{Pr + Im - Ex}$<br>In the equation: *Pr* represents the production of related products from the resources. *Im* represents the imports of related products from the resources. *Ex* represents the exports of related products from the resources. The trade data are sourced from the United Nations Commodity Trade database. |
| Market Risks (*FLS*) | Monopoly Level (*FLS1*) | - | $FLS1 = \sum Im_i$<br>In the equation: *i* represents a specific country. *Im* represents the import share of products (all types of refined tin). The trade data are sourced from the United Nations Commodity Trade database. |
| | Price Volatility (*FLS2*) | - | $FLS2 = \frac{Ps_n - Ps_{n-1}}{Ps_{n-1}}$<br>In the equation: *n* represents the year. *Ps* represents the import price of resources. The trade data are sourced from the United Nations Commodity Trade database. |
| Geopolitical Security (*FLD*) | Geopolitical Risk (*FLD1*) | + | $FLD1 = \sum WGI_i \times \frac{P_i}{P}$<br>In the equation: $WGI_i$ represents the World Governance Index of country *i*, specifically the parameter of political stability. $P_i$ represents the import quantity of mineral resources from country *i*. *P* represents the total import quantity of the mineral resource in question. The global governance index is sourced from the World Bank, and trade data are from the United Nations Comtrade. |
| | Supply Channel Security (*FLD2*) | + | $FLD2 = \sum CS_i \times \frac{Im_i}{Im}$<br>In the equation: $CS_i$ represents the safety index of route *i*. $Im_i$ represents the transportation volume of mineral resources relying on route *i*. *Im* represents the total import volume of mineral resources. The trade data are sourced from the United Nations Commodity Trade database. |

### 3.2. Research Methods

In view of the characteristics of diverse data types and the large data volume in the evaluation of sustainable supply capacity of China's tin resources, this study adopts the entropy weight method and TOPSIS (Technique for Order Preference by Similarity to Ideal Solution) model to evaluate the sustainable supply capacity of China's tin resources. The TOPSIS model quantitatively measures the distance between different data and the optimal values, accurately evaluating the safety level at different time points and determining the optimal solution among multiple alternatives [41–43]. Firstly, this study constructs a weighted matrix using the entropy weight method, and then applies the TOPSIS model to evaluate the supply security of China's tin resources at different time periods.

(1) Determination of the objective sequence: Since the indicators in the tin resource evaluation system have different initial dimensions, it is necessary to normalize the units of the indicators. In this study, the entropy weight method is employed to process the original data and achieve normalization.

$$y_{ij} = \frac{x_{ij} - min(x_{ij})}{max(x_{ij}) - min(x_{ij})} \text{ (for positive indicators)} \tag{1}$$

$$y_{ij} = \frac{max(x_{ij}) - x_{ij}}{max(x_{ij}) - min(x_{ij})} \text{ (for negative indicators)} \tag{2}$$

In the equation, *i* represents the evaluation indicator (i = 1, 2, 3, . . ., *m*) and *j* represents the indicator year (*j* = 1, 2, 3, . . ., *n*). $K_{ij}$ represents the initial value of the evaluation indicator system, $max(K_{ij})$ represents the maximum value of the indicator $K_{ij}$ and $min(K_{ij})$ represents the minimum value of the indicator $K_{ij}$. $x_{ij}$ represents the standardized value. The standardized matrix *B* is obtained through normalization.

$$B = \begin{bmatrix} x_{11} & \cdots & x_{1n} \\ \vdots & \ddots & \vdots \\ x_{m1} & \cdots & x_{mn} \end{bmatrix} \tag{3}$$

(2) Calculation of Entropy Weight. Firstly, the matrix *B* is normalized to obtain matrix $f_{ij}$. The calculation formula is as follows:

$$f_{ij} = \frac{x_{ij}}{\sum_{j=1}^{n} x_{ij}} \tag{4}$$

Next, we calculate the information entropy $e_i$ using the following formula:

$$e_i = -\frac{1}{\ln n} \sum_{j=1}^{n} f_{ij} \times \ln f_{ij} \tag{5}$$

If $f_{ij} = 0$, then:

$$\lim_{f_{ij} \to 0} f_{ij} \times \ln f_{ij} = 0 \tag{6}$$

Finally, the weights of the indicators $w_i$ are calculated using the following formula:

$$w_i = \frac{1 - e_i}{m - \sum_{i=0}^{m} e_i} \tag{7}$$

(3) Construction of TOPSIS Model: To ensure the objectivity of the evaluation results, a normalized analysis matrix *C* is created based on the indicator weights $w_i$.

$$C = \left| y_{ij} \right|_{m \times n} = \left| w_i \times x_{ij} \right|_{m \times n} \tag{8}$$

(4) Determination of Positive and Negative Ideal Values: The positive ideal solution $Z^+$ and the negative ideal solution $Z^-$ are determined based on the maximum and minimum values of the $i$-th indicator over the $j$ years. The specific formulas are as follows:

$$Z^+ = \max\{y_{ij}\}$$
$$Z^- = \min\{y_{ij}\}$$

(9)

(5) Calculation of Distances: The Euclidean distance is used to calculate the distance between the evaluation indicators of China's tin resource security and the positive and negative ideal solutions. $D^+$ represents the distance between the $i$-th indicator and $Z^+$, while $D^-$ represents the distance between the $i$-th indicator and $Z^-$. The specific formulas are as follows:

$$D^+ = \sqrt{\sum_{i=1}^{m}\left(Z_i^+ - y_{ij}\right)^2}$$
$$D^- = \sqrt{\sum_{i=1}^{m}\left(Z_i^- - y_{ij}\right)^2}$$

(10)

(6) Calculation of Comprehensive Evaluation Index: $C_j$ represents the comprehensive evaluation index of China's tin resource security for the $j$-th year. A higher value of $C_j$ indicates a higher level of security. The calculation formula is as follows:

$$C_j = \frac{D_j^-}{D_j^+ + D_j^-}$$

(11)

### 3.3. Data Sources

The tin production and reserve data used in this study are sourced from the United States Geological Survey (USGS). The import data of refined tin ore in China and global tin trade price data are obtained from the United Nations Commodity Trade Statistics Database. The Economic Freedom Index is derived from the Heritage Foundation, while the Global Governance Index is sourced from the World Bank. These data sources provide reliable and internationally recognized information for the analysis and evaluation of China's tin mining resources' sustainable supply capacity.

## 4. Indicator Analysis and Data Selection

(1) Supply Security

Supply security is a key aspect in evaluating the sustainable supply capacity of tin mineral resources. It is influenced by both domestic resource potential and the commercial mining environment abroad. The mining environment refers to the macro environment in which tin mining occurs in a foreign country or region. It reflects the ease or difficulty of the trade, finance, currency, investment, and labor aspects of tin mining. The mining environment is quantitatively represented by the product of the global share of tin production and the economic freedom index, with a higher value indicating a more favorable commercial environment for resource extraction.

Tin resources are distributed globally, with Asia having the highest concentration. Major tin reserves are found in Indonesia (800,000 tons), China (720,000 tons), and Myanmar (700,000 tons), collectively accounting for 48.1% of global tin reserves. Additionally, Australia (570,000 tons), Russia (430,000 tons), Brazil (420,000 tons), and Bolivia (400,000 tons) hold significant reserves. These seven countries account for 88% of the world's total tin reserves. According to the United States Geological Survey, China's tin reserves have gradually declined from 3.5 million tons in 2000 to 1.5 million tons in 2010 and further reduced to 720,000 tons in 2022. However, due to the rapid growth in production, China's tin reserve-to-production ratio has significantly decreased from 36.1 years in 2000 to 7.6 years in 2022 (Figure 2). This decline has led to a substantial decrease in China's advantage in tin resources, as the reserve-to-production ratio currently stands at 51.1% of the global average.

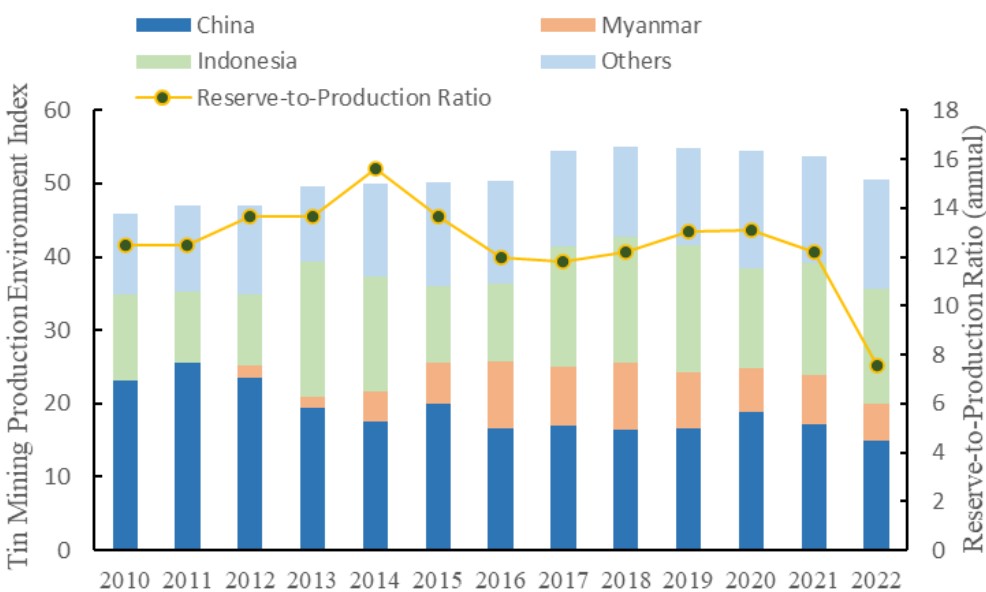

**Figure 2.** Tin Mining Environment Index and China's Reserve-to-Production Ratio.

(2) Demand Situation

The demand situation is influenced by the growth rate of resource consumption and the degree of external dependence on resources. In terms of the relevant raw materials and primary products of tin, China has been a net importer of tin ore and concentrates since 2010. The net import quantity is calculated by converting related products into tin concentrates, and the apparent consumption is calculated based on China's tin concentrate production. According to the statistical data (Table 2), China's net import of tin resources has increased from 19,800 tons in 2010 to 244,000 tons in 2022, with an average annual growth rate of 12.1%. The apparent consumption has increased from 130,000 tons in 2010 to 217,000 tons in 2022, showing a growth of 66.9% and an average annual growth rate of 2.4%. From 2010 to 2022, China's external dependence on tin resources has rapidly risen from 7.6% to 56.2%, reaching its peak of 61.4% in 2017.

**Table 2.** Evaluation Index System for Sustainable Development of China's Tin Resources. Unit: 10,000 metric tons.

| Year | 2010 | 2011 | 2012 | 2013 | 2014 | 2015 | 2016 | 2017 | 2018 | 2019 | 2020 | 2021 | 2022 |
|---|---|---|---|---|---|---|---|---|---|---|---|---|---|
| Import Volume | 2 | 3 | 3 | 10 | 18 | 29 | 26 | 30 | 22 | 18 | 16 | 18 | 24 |
| Export Volume | 0 | 0 | 0 | 0 | 0 | 0 | 0 | 0 | 0 | 0 | 0 | 0 | 0 |
| Production Volume | 12 | 12 | 11 | 11 | 9.6 | 11 | 9.2 | 9.3 | 9 | 8.45 | 8.4 | 9 | 9.5 |
| Net Import Volume | 1.0 | 1.4 | 1.6 | 4.8 | 8.9 | 14.6 | 13.0 | 14.8 | 11.1 | 8.9 | 7.9 | 9.2 | 12.2 |
| Apparent Consumption Volume | 13.0 | 13.4 | 12.6 | 15.8 | 18.5 | 25.6 | 22.2 | 24.1 | 20.1 | 17.4 | 16.3 | 18.2 | 21.7 |
| Import Volume | 7.6% | 10.7% | 12.8% | 30.5% | 48.1% | 57.0% | 58.6% | 61.4% | 55.2% | 51.3% | 48.5% | 50.6% | 56.2% |

(3) Market Risk

Market risk primarily refers to the degree of global resource monopolization and the magnitude of price fluctuations. The degree of resource monopolization is reflected based on the monopoly situation in the countries that are the sources of resource imports. Analyzing the trade situation of tin ore in China (Figure 3), it can be observed that the import sources of tin ore are relatively concentrated. Among the main import sources, Myanmar accounts for 76.9% of the total import volume, while imports from countries such as the Democratic Republic of Congo and Australia have shown some growth in recent years. The import prices of tin ore have experienced significant fluctuations over the years, starting from the lowest point of $1282 per ton in 2015 and reaching the highest point of $8374 per ton in 2022.

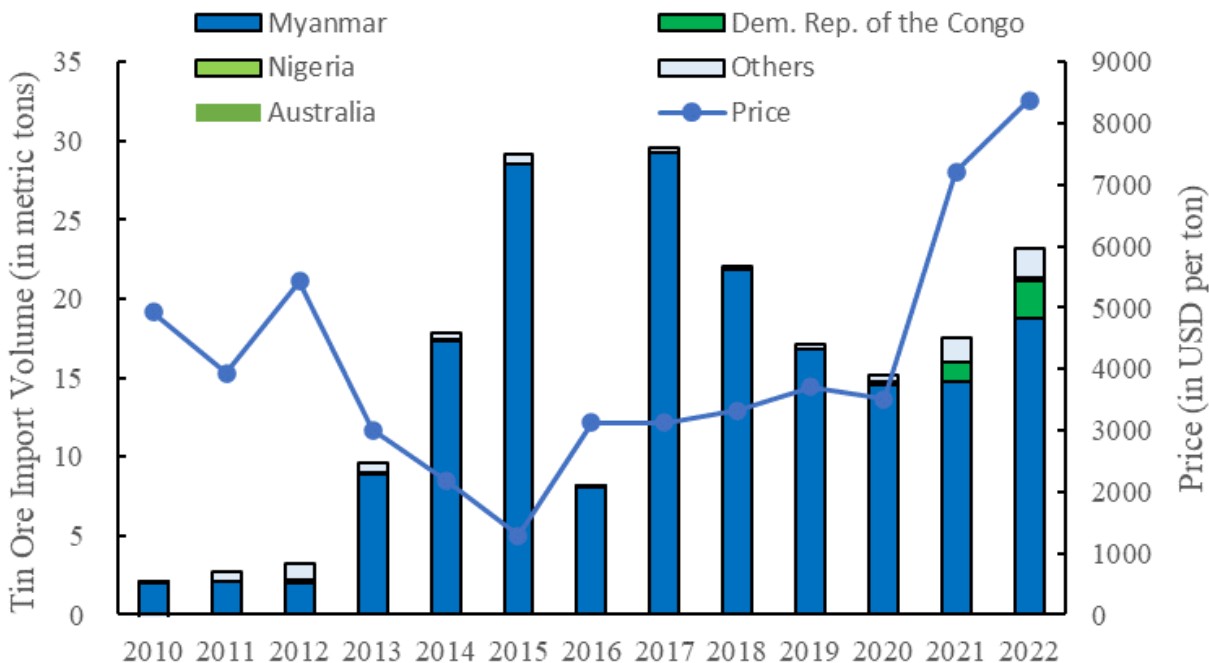

**Figure 3.** Importing Countries and Price Variation of Chinese Tin Resources.

(4) Geopolitical Security

Geopolitical security is influenced by the stability of resource supply nations and the security level of resource import channels. In this study, the Geopolitical Risk Index is constructed by multiplying the tin ore import volume with the corresponding Global Governance Index of the countries. The Channel Security Index is formulated following the approach of Li, J. (2017) [44] and Xing, L. (2022) [45], wherein China's maritime channels are categorized as the Western Pacific, Eastern Pacific, European, Australian, and Americas–Africa channels. The safety values are quantified by considering various risk factors, including natural risks, countries along the route, and military control. Based on the calculated results (Figure 4), the overall Geopolitical Risk Index for China's tin resources demonstrates an oscillating declining trend. It decreases from −3.9 in 2010 to −39.0 in 2015, rises to −7.8 in 2016, and then slightly recedes to −34.9 in 2022.

The Channel Security Index for the supply pathways of China's tin resources exhibits a symmetrical trend with the Geopolitical Risk Index. It rises from 2.0 in 2010 to 28.6 in 2015, drops sharply to 8.1 in 2016, rebounds to 29.2 in 2017, and gradually declines to the current value of 16.7. The geopolitical risk and channel security of China's tin resources are influenced by the import volume and sources. The import volume increases from 2.0 thousand tons in 2010 to 29.5 thousand tons in 2017, then decreases to 24.4 thousand tons in 2022. The significant drop in the import volume in 2016 to 8.2 thousand tons, representing a 71.9% reduction compared to 2015, was the primary cause of the fluctuation in the geopolitical security index in 2015.

China, starting from 2010, imported nearly 99% of its tin resources from Myanmar. However, from 2019 onwards, there has been an increase in tin ore imports from the African Congo (Gold) and Australia, diversifying the sources and thereby enhancing the security of China's tin resource supply channels. Despite the influence of geopolitical relations, there is a declining trend in the geopolitical risk of China's tin resources.

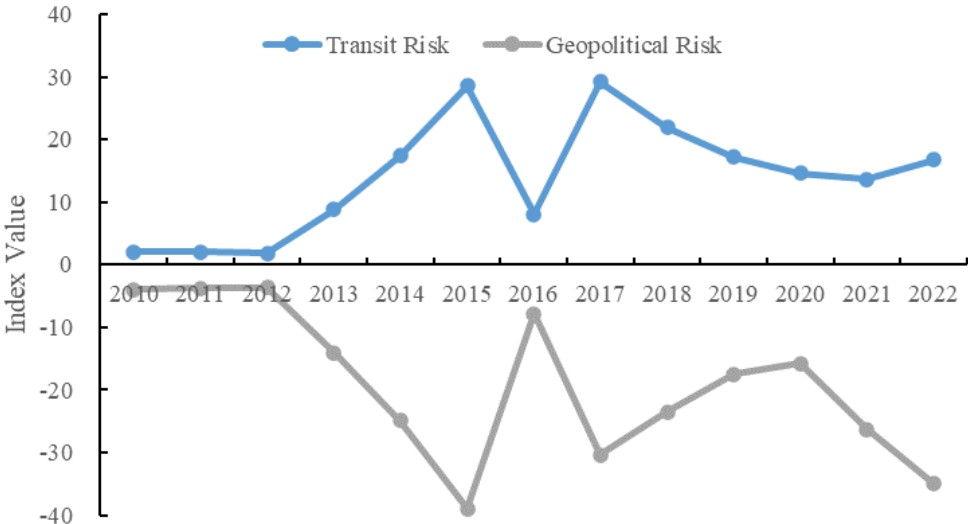

**Figure 4.** Geopolitical and Supply Channel Risk Indices for China's Tin Resource (2010–2022).

## 5. Results and Analysis

Based on the indicator system for evaluating the security of China's tin resource, constructed according to Table 1, six experts in the field were invited to individually rate the importance of each indicator. Through internal discussions and summarization of the ratings, equal weight was attributed to each expert's opinion. Using the aforementioned ratings, we employed the Analytic Hierarchy Process (AHP) to calculate the weights of the indicators, and the eigenvector method was employed to derive the weights, as detailed in Table 3. Additionally, by combining the eigenvector, we calculated the maximum eigenvalue to be 8.685. Subsequently, using this maximum eigenvalue, the Consistency Index (CI) was computed to be 0.098, indicating satisfactory consistency within the judgment matrix.

**Table 3.** Results of Indicator Weight Analysis Based on the Analytic Hierarchy Process (AHP).

| Secondary Indicator | Eigenvector | Weight Values | Maximum Eigenvalue | CI Value |
|---|---|---|---|---|
| Business Environment | 0.266 | 0.033 | | |
| Domestic Supply Potential | 2.435 | 0.304 | | |
| Consumption Growth Rate | 0.202 | 0.025 | | |
| External Dependence | 1.644 | 0.206 | 8.685 | 0.098 |
| Monopoly Degree | 0.574 | 0.072 | | |
| Price Fluctuation | 0.399 | 0.050 | | |
| Geopolitical Risk | 1.12 | 0.140 | | |
| Supply Channel Security | 1.361 | 0.170 | | |

Based on the entropy weighting method, the weights of the evaluation indicators were calculated, as shown in Table 4. The importance of demand situation, supply security, geopolitical security, and market risk in ensuring the supply security of China's tin resources decreases in that order. Among them, domestic supply potential (0.274), external dependency (0.213), and supply channel security (0.154) are the most critical factors influencing the supply security of China's tin resources.

Using the entropy weight method and TOPSIS model, a comprehensive evaluation of China's tin resource supply security was conducted, and the evaluation results are presented in Table 5. Based on this, the overall changes in China's tin resource supply security can be identified, and the key factors influencing China's tin resource supply security can be clearly understood when combined with the analysis of China's tin resource supply situation.

**Table 4.** Weights of Sustainable Supply Indicators for China's Tin Resources.

| Primary Indicator | Secondary Indicator | Entropy Weight | Entropy Weight | Entropy Weight |
|---|---|---|---|---|
| Supply Security (FLG) | Business Environment | 0.054 | 0.033 | 0.044 |
| | Domestic Supply Potential | 0.243 | 0.304 | 0.274 |
| Demand Situation (FLX) | Consumption Growth Rate | 0.052 | 0.025 | 0.039 |
| | External Dependence | 0.220 | 0.206 | 0.213 |
| Market Risk (FLS) | Monopoly Degree | 0.083 | 0.072 | 0.078 |
| | Price Fluctuation | 0.076 | 0.050 | 0.063 |
| Geopolitical Security (FLD) | Geopolitical Risk | 0.133 | 0.140 | 0.136 |
| | Supply Channel Security | 0.139 | 0.170 | 0.154 |

**Table 5.** Evaluation Results of China's Tin Resource Sustainable Supply Capacity.

| Year | Indicator Score | | | | Euclidean Distance | | Sustainability Index |
|---|---|---|---|---|---|---|---|
| | FLG | FLX | FLS | FLD | $D^+$ | $D^-$ | |
| 2010 | 0.167 | 0.237 | 0.036 | 0.137 | 0.039 | 0.091 | 0.577 |
| 2011 | 0.173 | 0.224 | 0.114 | 0.138 | 0.031 | 0.092 | 0.649 |
| 2012 | 0.207 | 0.219 | 0.113 | 0.136 | 0.028 | 0.101 | 0.674 |
| 2013 | 0.211 | 0.139 | 0.079 | 0.136 | 0.032 | 0.066 | 0.565 |
| 2014 | 0.282 | 0.072 | 0.063 | 0.142 | 0.046 | 0.077 | 0.559 |
| 2015 | 0.209 | 0.030 | 0.066 | 0.151 | 0.072 | 0.058 | 0.456 |
| 2016 | 0.151 | 0.160 | 0.002 | 0.156 | 0.048 | 0.051 | 0.470 |
| 2017 | 0.176 | 0.000 | 0.050 | 0.188 | 0.082 | 0.042 | 0.414 |
| 2018 | 0.078 | 0.054 | 0.049 | 0.173 | 0.063 | 0.041 | 0.354 |
| 2019 | 0.225 | 0.068 | 0.057 | 0.170 | 0.050 | 0.048 | 0.520 |
| 2020 | 0.231 | 0.077 | 0.067 | 0.162 | 0.046 | 0.050 | 0.537 |
| 2021 | 0.201 | 0.064 | 0.055 | 0.116 | 0.061 | 0.033 | 0.436 |
| 2022 | 0.020 | 0.039 | 0.091 | 0.099 | 0.120 | 0.013 | 0.249 |

Based on Figure 5, it can be observed that the score of China's tin resource security evaluation indicators has decreased from 0.578 in 2000 to 0.249 in 2022, indicating a continuous decline in the level of tin resource supply security in China since 2010. Since 2000, the supply security of China's tin resources has experienced an initial increase followed by a decline, while the demand situation initially declined and then showed some recovery. Geopolitical security remained relatively stable, and market risks fluctuated before decreasing and then rebounding. The supply security of tin resources in China has exhibited an overall downward trend, following an M-shaped pattern from 2000 to 2022. This trend is mainly attributed to the deterioration of the international mining environment and the reduction in domestic resource supply potential. The decline in the global proportion of tin production from countries such as Myanmar and Indonesia, coupled with the increasing production of tin resources in countries like the Democratic Republic of Congo in Africa, has led to a gradual deterioration of the security environment for tin mining. The significant decrease in domestic resource potential, with the time to depletion decreasing from 36.1 years in 2000 to 7.6 years in 2022, corresponds to the current level of 51.1% compared to the international average, indicating that China's tin resources no longer hold a competitive advantage. The index for the demand situation of China's tin resources has declined from 0.237 in 2010 to 0.030 in 2015, and then experienced fluctuations with a slight recovery to around 0.046. The annual growth rate of tin consumption in China has shown fluctuations, with significant growth observed in 2013, 2015, 2017, and 2022, with growth rates of 25.4%, 38.2%, 81.3%, and 19.0%, respectively. Future resource consumption demands are expected to increase rapidly, indicating a challenging outlook for the demand situation of tin resources.

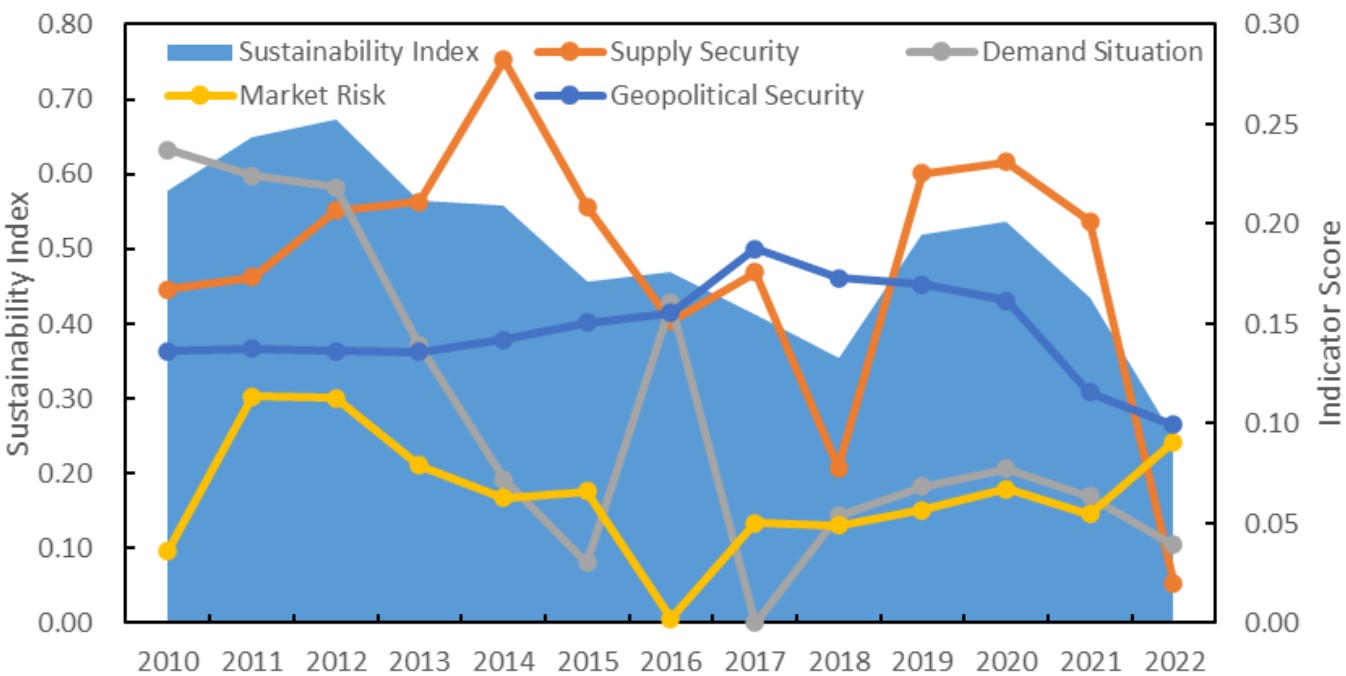

**Figure 5.** Trends in the Sustainable Supply Capacity of Chinese Tin Resources.

The market risk of China's tin resources has experienced a slight increase. Firstly, the rapid price surge since 2020 has heightened market risk. Furthermore, the import sources of Chinese tin concentrates have gradually diversified, with Myanmar's proportion as the largest import country declining from 99.7% in 2015 to 76.9% in 2022. The diversified supply system has enhanced the security of China's tin resources.

In terms of geopolitical security, China's imports from Africa have been gradually increasing, resulting in a significant decrease in the shipping distance, security level, and political stability of resource-supplying countries compared to Myanmar and Australia.

Figure 6 presents a radar chart illustrating various indicators of China's tin mining resource sustainable supply capacity. A higher index value indicates a higher level of safety for the respective indicator. Notably, it is evident that market risk was mitigated by 2010, while other key factors experienced significant deterioration. Specifically, the indices for external dependency, geopolitical risk, supply security, and demand situation decreased by 90.3%, 88.5%, 83.4%, and 81.6%, respectively. At present, China's tin resource reserves have undergone a substantial decline, leading to a continuous reduction in safeguard years. The reliance on foreign-sourced tin raw materials remains considerable. With the development of China's high-end manufacturing and strategic emerging industries, the demand for tin metal is anticipated to continue growing, thereby imposing long-term pressure on resource supply. Furthermore, the importation of tin resources from Africa has gradually expanded, introducing adverse effects on the political stability of supplying nations and maritime route safety.

Overall, the outlook for the supply security of China's tin resources is not optimistic. Geopolitical risks, supply security, and demand conditions have shown noticeable declines. Weak military safeguards for resource transportation channels, poor political stability of import source countries, and rapid depletion of domestic resource reserves have become the major factors limiting the security of China's tin resources.

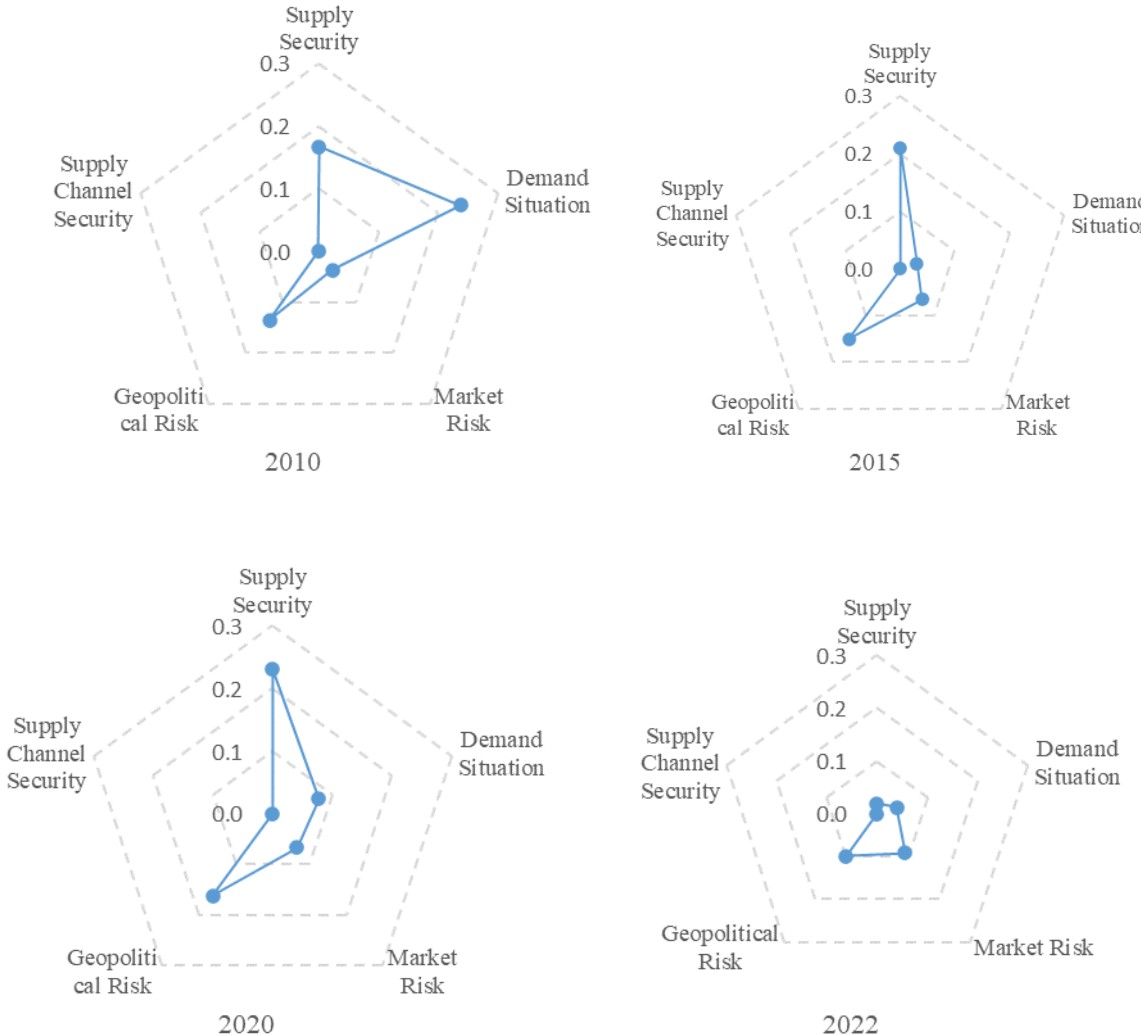

**Figure 6.** Radar Chart of Sustainable Capacity for Chinese Tin Resources.

## 6. Conclusions and Recommendations

Based on the comprehensive analysis and evaluation of the sustainable supply capacity of Chinese tin resources from the aspects of supply security, demand situation, market risks, and geopolitical security, the following conclusions have been drawn in this study. The main conclusions of this paper are consistent with the research findings of other scholars such as Li, H. (2021) [1], Yang, C. (2018) [3], Wanli, X. (2016) [9], Wang, J. (2018) [17], Zhai, M. (2021) [34], and Fan, Z. (2022) [40].

### 6.1. Conclusions and Recommendations

(1) The level of supply security for Chinese tin resources has shown a continuous decline. The results of the comprehensive evaluation using the entropy weight method and TOPSIS model indicate an overall downward trend in the supply security of Chinese tin resources since 2000. This decline can be attributed to the deteriorating international mining environment, decreasing domestic resource potential, and changes in the global tin mining landscape. Therefore, measures need to be taken to enhance the supply security of Chinese tin resources.

(2) The domestic supply potential of tin resources in China has declined. There has been a significant decrease in the domestic tin resource potential in China, with the extractable years declining from 36.1 years in 2000 to 7.6 years in 2022, representing only 51.1% of the international average. This indicates that Chinese tin resources have lost their

advantage. To ensure supply security, measures should be implemented to improve the efficiency of exploration and development of domestic tin resources.

(3) The outlook for tin resource demand is not optimistic. The demand situation index for Chinese tin resources has decreased from 0.237 in 2010 to 0.030 in 2015 but has shown some recovery thereafter. Future consumption demand for resources is expected to grow rapidly, posing pressure on the supply security of tin resources. Therefore, it is necessary to formulate sound resource management policies, promote resource conservation and utilization, and drive green manufacturing and sustainable development.

(4) Geopolitical challenges are faced in the security of tin resources. With the increasing proportion of tin imports from Africa, geopolitical risks and maritime security issues have become more prominent. Additionally, the political stability of the source countries also affects supply security. To mitigate geopolitical security risks, diversification of import channels, strengthening cooperation with supplier countries, and enhancing maritime security capabilities are recommended.

In conclusion, to ensure the sustainable supply security of Chinese tin resources, the following measures are recommended. The first is to strengthen domestic tin resource exploration and development, improve resource extraction efficiency, and reduce resource wastage. Second, to formulate and implement science-based resource management policies and enhance resource conservation and recycling of tin resources. Third, to promote green manufacturing and sustainable development, reduce dependence on tin resources, and optimize industrial structure and transformation. Fourth, to diversify import channels, reduce reliance on specific countries or regions, and minimize the impact of geopolitical risks on supply security. Finally, to establish a robust maritime security system and enhance military safeguarding capabilities for tin resource transportation channels.

*6.2. Limitations and Prospects*

The connotation of mineral resource security undergoes dynamic changes with the evolution of eras. Presently, the global geopolitical landscape, economic patterns, technological forms, and energy structures are all experiencing profound transformations. In the future, as the international competitive landscape evolves and China's comprehensive strength continues to grow, it will be imperative to reexamine the factors influencing the sustainable supply security of China's tin resources. Due to professional limitations, this paper provides only a concise analysis of the framework and logic of the evaluation system for China's sustainable tin resource supply. A more comprehensive investigation into its operational status, mechanism essence, and logical structure from the perspectives of international relations, macroeconomics, and other professional viewpoints is essential. As the global context evolves, an in-depth study from these perspectives will contribute to a deeper understanding of the subject matter.

**Author Contributions:** Conceptualization, X.F.; data curation, Q.Y.; formal analysis, J.X.; methodology, X.F. and Q.Y.; resources, J.X.; validation, H.L. and J.X.; visualization, X.F.; writing—original draft, H.L.; writing—review and editing, Q.Y. and M.L. All authors have read and agreed to the published version of the manuscript.

**Funding:** This research was funded by the China Geological Survey Project (No. KZ11Z250).

**Institutional Review Board Statement:** Not applicable.

**Informed Consent Statement:** Not applicable.

**Data Availability Statement:** Not applicable.

**Conflicts of Interest:** The authors declare no conflict of interest.

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
