# Peer review of "Assessment of Sustainable Supply Capability of Chinese Tin Resources Based on the Entropy Weight-TOPSIS Model"

_sustainability, doi:10.3390/su151713076_

Round 1
Reviewer 1 Report
Introduction - While there is a good introduction of what tin is and the sustainable mineral extraction literature as a whole, we don't know what makes tin specifically important. "...rapid extraction and excessive utilization, leading to threats of resource depletion, environmental degradation, and social issues in many mining areas..." applies to a lot of minerals. It's also crucial to provide some insight of tin specific literature.
Global Resources - The pie charts don't really serve any function and distracts from the process flow diagram. Sure, Chinese usage of tin is slightly different than global usage, but what does that have to do with the process flow diagram?
Indicator system - Isn't geopolitical risk a part of resource security given that you mention international markets in the latter? Specifically it seems that FLG1 is mostly concerned with the domestic market and the similar FLD1 is concerned with the international market. This needs to be better clarified.
Indicator analysis and data selection - Table 3 needs better explanation of units
Results - Am I interpreting the radar charts right in the sense that they show in each year what are the major factors contributing to that years sustainability score, e.g. in 2022 - market risk plays the biggest role
Author Response
Thank you for your comments sincerely, which are very important for this study, and I will respond to your suggestions below.
Point 1: Introduction - While there is a good introduction of what tin is and the sustainable mineral extraction literature as a whole, we don't know what makes tin specifically important. "...rapid extraction and excessive utilization, leading to threats of resource depletion, environmental degradation, and social issues in many mining areas..." applies to a lot of minerals. It's also crucial to provide some insight of tin specific literature.
Response 1:
Thank you for your suggestions. In fact, the background of this paper can be summarized into three points: firstly, with the adjustment of China's industrial structure, the future demand for tin ore resources is expected to increase; secondly, excessive exploitation of tin ore resources in China has led to a decline in resource extraction security; and thirdly, the intensifying global geopolitical competition has impacted China's overseas resource supply chain. We have made additional clarifications in the introduction section, particularly on the third point (Lines 33-488).
Point 2: Global Resources - The pie charts don't really serve any function and distracts from the process flow diagram. Sure, Chinese usage of tin is slightly different than global usage, but what does that have to do with the process flow diagram?
Response 2:
Section where Figure 1 is situated primarily serves to provide readers with an overview of the tin ore resource industry chain, including resource characteristics, consumption structure, trade patterns, and demand trends. Its main purpose is to lay the foundation for the subsequent content. Figure 1 mainly illustrates the material flow of tin ore resources, and the initial version did not include the pie chart. However, after considering that a simple material flow diagram couldn't adequately demonstrate the consumption structure of tin ore resources, we added the consumption structure pie chart in the second version (Lines 175-176).
Point 3: Indicator system - Isn't geopolitical risk a part of resource security given that you mention international markets in the latter? Specifically it seems that FLG1 is mostly concerned with the domestic market and the similar FLD1 is concerned with the international market. This needs to be better clarified.
Response 3:
Thank you for your suggestion. This part should be a clarification of a point not clearly articulated in the article. In line 214, where it mentions 'Resource security,' it should be replaced with 'Supply Security.' In reality, tin ore resource security is a broad concept, encompassing supply security (FLG), demand trends (FLX), market risk (FLS), and geopolitical security (FLD) in this study. Among these, supply security mainly focuses on the stability of international supply (FLG1) and domestic supply (FLG2). The stability of international supply (FLG1) is primarily influenced by commercial activities, so the calculation uses the Economic Freedom Index of various countries to assess the business environment. On the other hand, geopolitical security considers the geopolitical risks of countries that are sources of international supply (FLD1) and the risks associated with supply channels (FLD2). Geopolitical risk (FLD1) is mainly evaluated using the Global Governance Index (GGI) (Lines 214).
Point 4: Indicator analysis and data selection - Table 3 needs better explanation of units
Response 4:
Thank you for your suggestion. We have carefully reviewed this section and have decided to rewrite it. (Lines 319-344).
Point 5: Results - Am I interpreting the radar charts right in the sense that they show in each year what are the major factors contributing to that years sustainability score, e.g. in 2022 - market risk plays the biggest role
Response 5:
Thank you for your suggestion. We have made the necessary adjustments to the figures and have rewritten the content in this section. (Lines 406-420).

Reviewer 2 Report
Evaluation of Sustainable Development Capacity of Tin Min- 2 eral Resources in China Based on Entropy Weight-TOPSIS 3 Model
Major comments
Please make sure the abstract and introduction are clear including an economic argument as to why you found something rather than simply saying what you have found from your analysis, highlighting its novelty relative to the literature.
The research gap and contribution of the study should be highlighted in the abstract. The methodology used in this research should be fully presented in the abstract.
The source of Figure 1 should be presented.
The time trend of the main indicators of the research should be presented in the form of a diagram.
Provide sources for the indicators presented in Table 1.
Is the attributes of the Geopolitical Risk Index positive?
Sustainability indicators should be considered when evaluating sustainable development. While environmental indicators are not taken into account in this study. This is one of the most serious shortcomings of this study. In any case, because environmental indicators are not taken into account, it is recommended that these indicators be included in the analysis or that the term "sustainable development" be renamed.
For the TOPSIS method section, use the references for its equations. You can use the following articles in this regard.
Balcerzak, A. P., & Pietrzak, M. B. (2016). Application of TOPSIS method for analysis of sustainable development in European Union countries (No. 22/2016). Institute of Economic Research Working Papers.
Radmehr, R., & Shayanmehr, S. (2018). The determinants of sustainable irrigation water prices in Iran. Bulgarian Journal of Agricultural Science, 24(6).
Because the entropy method is a statistical method, its results do not always correspond to reality. It is recommended that the weights be calculated using AHP and used in TOPSIS experiments and analysis to ensure the robustness of the results
The discussion is well written, but the authors should like their findings to the previous studies in the literature.
The results of the sensitivity analysis need to be presented.
There is the need for professional proofreading or consulting English native support
The policy recommendations of the study are weak and small. I, therefore, encourage the authors to elaborate more on the policy recommendations to policymakers for achieving sustainable development.
The authors should add limitations of the study and future recommendation
There is a need for professional proofreading or consulting with native English support
Author Response
Thank you for your comments sincerely, which are very important for this study, and I will respond to your suggestions below.
Point 1: Please make sure the abstract and introduction are clear including an economic argument as to why you found something rather than simply saying what you have found from your analysis, highlighting its novelty relative to the literature. The research gap and contribution of the study should be highlighted in the abstract. The methodology used in this research should be fully presented in the abstract.
Response 1:
Thank you for your suggestions. The background of this study can be summarized into three points: First, with the adjustment of China's industrial structure, the demand for tin resources is expected to increase in the future. Second, excessive exploitation of tin resources in China has led to a decline in the degree of resource extraction security. Third, intensified international geopolitical competition has impacted China's overseas resource supply chain. In order to make the content of the abstract and introduction clearer, we have rewritten the abstract and made adjustments to the introduction. (Lines 14-28, Lines 33-488, Lines 124-135).
Point 2: The source of Figure 1 should be presented.
Response 2:
Figure 1 is original to this study, and the process files have been submitted to the editor via email.
Point 3: The time trend of the main indicators of the research should be presented in the form of a diagram.
Response 3:
Thank you for your suggestion. Section 4, Indicator Analysis and Data Selection, specifically presents the time trends of various indicator data.
Point 4: Provide sources for the indicators presented in Table 1.
Response 4:
Thank you for your suggestion. We have added the data sources for each indicator in the section explaining the calculation formulas. (Lines 26-27)
Point 5: Is the attributes of the Geopolitical Risk Index positive?
Response 5:
Yes, the attribute of the geopolitical risk indicator is positive. In this study, geopolitical risk is calculated as the product of tin ore import volume and the global governance index of the respective country. A higher global governance index indicates greater stability in that country, implying a lower geopolitical risk. (Lines 26-27)
Point 6: Sustainability indicators should be considered when evaluating sustainable development. While environmental indicators are not taken into account in this study. This is one of the most serious shortcomings of this study. In any case, because environmental indicators are not taken into account, it is recommended that these indicators be included in the analysis or that the term "sustainable development" be renamed.
Response 6:
Thank you for your suggestion. After thorough consideration, we have revised the term 'sustainable development' to 'sustainable supply capacity' and made corresponding adjustments to both the title and the content within the paper.
Point 7: For the TOPSIS method section, use the references for its equations. You can use the following articles in this regard.
Response 7:
Thank you for your suggestion. We have included these two references in the paper (Lines 234).
Point 8: Because the entropy method is a statistical method, its results do not always correspond to reality. It is recommended that the weights be calculated using AHP and used in TOPSIS experiments and analysis to ensure the robustness of the results
Response 8:
Thank you for your suggestion. We consulted 6 experts in the field of resource economics, and based on their ratings, we calculated the weights of various indicators using the Analytic Hierarchy Process (AHP) method, and further combined it with the Entropy Weight Method to calculate the comprehensive weights of each indicator. The content of the paper has also been revised accordingly (Lines 349-369).
Point 9: The discussion is well written, but the authors should like their findings to the previous studies in the literature.
Response 9:
Thank you for your suggestion. We have compared the main conclusions with the results from existing literature, and the main viewpoints are generally consistent (Lines 432-435).
Point 10: There is the need for professional proofreading or consulting English native support
Response 10:
Thank you for your suggestions. We have reorganized the paper accordingly.
Point 11: The policy recommendations of the study are weak and small. I, therefore, encourage the authors to elaborate more on the policy recommendations to policymakers for achieving sustainable development.
Response 11:
Thank you for your suggestions. The perspective of this paper is focused on macro-level strategic research, and we haven't delved into finer details. However, we acknowledge the importance of conducting in-depth research on strategic minerals such as tin. We are currently or soon planning to investigate topics such as resource demand, resource supply patterns, resource markets, resource security, and global governance.
Point 12: The authors should add limitations of the study and future recommendation
Response 12:
Thank you for your suggestions. We have added a section titled '6.2 Limitations and Prospects' as per your recommendation (Lines 472-484).

Reviewer 3 Report
-Figure 1: how was this diagram prepared? There is lack of references and explanation.
-Table 1: please add a brief discussion in text regarding the various indexes
-Line 257: upply-> correct to supply
-Lines 425-434 are too generic (seem AI generated). Please make more relevant to the data considered in the paper
-Discuss the limitation of the model used
-Discuss the limitation of the data sources used
Minor grammar checks and correction needed.
Author Response
Response to Reviewer 3 Comments
Thank you for your comments sincerely, which are very important for this study, and I will respond to your suggestions below.
Point 1: Figure 1: how was this diagram prepared? There is lack of references and explanation.
Response 1:
Thank you for your suggestions. In this section, we have included image captions and provided the data sources. (Lines 138-146).
Point 2: Table 1: please add a brief discussion in text regarding the various indexes
Response 2:
Thank you for your feedback. In fact, the paper includes explanations of indicators in three instances. Firstly, the explanation of indicator selection in Table 1 elaborates on the interpretation of primary indicators. Secondly, within Table 1, each indicator's calculation method and data sources are explained. Thirdly, in the '4. Indicator Analysis and Data Selection' section of the paper, the significance of each indicator is explained in each subsection.
Point 3: Line 257: upply-> correct to supply
Response 3:
Thank you, this was an error that shouldn't have occurred. We have already corrected it.
Point 4: Lines 425-434 are too generic (seem AI generated). Please make more relevant to the data considered in the paper
Response 4:
Thank you for your suggestions. It's important to note that the perspective of our article is focused on macro-level strategic research, which might make the recommendations seem less specific. In reality, it is indeed crucial to study strategic minerals like tin ores. We are currently or will soon be delving into in-depth research on topics such as resource demands, resource supply patterns, resource markets, resource security, global governance, and more. We believe that future research will delve deeper and become more specific, addressing each intricate issue.
Point 5: Discuss the limitation of the model used. Discuss the limitation of the data sources used
Response 5:
Thank you for your suggestions. We have included a section titled '6.2 Limitations and Prospects. (Lines 478-490).

Round 2
Reviewer 2 Report
All comments have been well addressed.
Reviewer 3 Report
I am satisfied with the new updated version. Thank you.